# Competition for fluctuating resources reproduces statistics of species abundance over time across wide-ranging microbiotas

Po-Yi Ho[1], Benjamin H Good[2,3]*, Kerwyn Casey Huang[1,3,4]*

[1]Department of Bioengineering, Stanford University, Stanford, United States; [2]Department of Applied Physics, Stanford University, Stanford, United States; [3]Chan Zuckerberg Biohub, San Francisco, United States; [4]Department of Microbiology and Immunology, Stanford University School of Medicine, Stanford, United States

**Abstract** Across diverse microbiotas, species abundances vary in time with distinctive statistical behaviors that appear to generalize across hosts, but the origins and implications of these patterns remain unclear. Here, we show that many of these macroecological patterns can be quantitatively recapitulated by a simple class of consumer-resource models, in which the metabolic capabilities of different species are randomly drawn from a common statistical distribution. Our model parametrizes the consumer-resource properties of a community using only a small number of global parameters, including the total number of resources, typical resource fluctuations over time, and the average overlap in resource-consumption profiles across species. We show that variation in these macroscopic parameters strongly affects the time series statistics generated by the model, and we identify specific sets of global parameters that can recapitulate macroecological patterns across wide-ranging microbiotas, including the human gut, saliva, and vagina, as well as mouse gut and rice, without needing to specify microscopic details of resource consumption. These findings suggest that resource competition may be a dominant driver of community dynamics. Our work unifies numerous time series patterns under a simple model, and provides an accessible framework to infer macroscopic parameters of effective resource competition from longitudinal studies of microbial communities.

*For correspondence:
bhgood@stanford.edu (BHG);
kchuang@stanford.edu (KCH)

**Competing interest:** The authors declare that no competing interests exist.

## Editor's evaluation

This paper introduces an elegant mathematical and ecological framework to model the fluctuations of microbial abundances in microbiomes along time series. The modeling approach considers consumer-resource properties and is regulated by few parameters. Applied to time-series microbiome data the model suggests the existence of recurrent patterns of microbial dynamics that are quite dependent on resource competition.

## Introduction

Microbial communities are ubiquitous across our planet, and strongly affect host and environmental health (*Sekirov et al., 2010*; *Tkacz and Poole, 2015*). Predictive models of microbial community dynamics would accelerate efforts to engineer microbial communities for societal benefits. A promising class of models is consumer-resource (CR) models, wherein species growth is determined by the consumption of environmental resources (*Chesson, 1990*). CR models capture a core set of

interactions among members of a community based on their competition for nutrients, and have demonstrated the capacity to recapitulate important properties of microbial communities such as diversity and stability (*Niehaus et al., 2019*; *Posfai et al., 2017*; *Tikhonov and Monasson, 2017*). However, while model parameters such as resource consumption rates are beginning to be uncovered in the context of in vitro experiments (*Goldford et al., 2018*; *Hart et al., 2019*; *Liao et al., 2020*), it remains challenging to determine all parameters for a community of native complexity from the bottom-up. A more accessible approach to parametrize CR models and to understand the features that drive community-level properties is needed.

To interrogate the dynamics of in vivo microbiotas, a common, top-down strategy is longitudinal sampling followed by 16S amplicon or metagenomic sequencing, thereby generating a relative abundance time series. Analyses of longitudinal data have shown that species abundances fluctuate around stable, host-specific values in healthy humans (*Caporaso et al., 2011*; *David et al., 2014*; *Faith et al., 2013*). Recently, it was discovered that such time series exhibit distinctive statistical signatures, sometimes referred to as macroecological dynamics, that can reflect the properties of the community and its environment (*Descheemaeker and de Buyl, 2020*; *Grilli, 2020*; *Ji et al., 2020*; *Shoemaker et al., 2017*). For example, in human and mouse gut microbiotas, the temporal variance of different species scales as a power of their mean abundance ('Taylor's law', *Taylor, 1961*) and deviations from this trend can highlight species that are transient invaders (*Ji et al., 2020*). Time series modeling can also provide insights into the underlying ecological processes. For example, the relative contributions of intrinsic versus environmental processes can be distinguished using autoregressive models whose output values depend linearly on values at previous times and external noise (*Gibbons et al., 2017*). Time series can also be correlated to environmental metadata such as diet to generate hypotheses about how environmental perturbations affect community composition (*David et al., 2014*), and to identify environmental drivers of transitions between distinct ecological states (*Levy et al., 2020*).

A growing body of work has shown that time series generated by simple mathematical models can exhibit statistics similar to experimental data sets, reinforcing the utility of such models for providing information about community dynamics even when many microscopic details are unknown. Some statistics can be recapitulated by phenomenological models, such as a non-interacting, constrained random walk in abundances (*Grilli, 2020*), while others can be described by a generalized Lotka-Volterra (gLV) model with colored noise (*Descheemaeker and de Buyl, 2020*) or by ecological models describing the birth, immigration, and death of species (*Azaele et al., 2006*). However, the origins of and relationships among time series statistics have yet to be explained. Here, we sought to address this question using CR models, and simultaneously to use time series statistics as an accessible approach for parametrizing CR models.

Since the network of resource consumption in a community will typically depend on thousands of underlying parameters, directly measuring all parameters is intrinsically challenging. We sought to overcome this combinatorial complexity by adopting an indirect, coarse-grained approach, in which resources describe effective groupings of metabolites or niches, and model parameters are randomly drawn from a common statistical ensemble. We show that this simple formulation generates statistics that quantitatively match those observed in experimental time series across wide-ranging microbiotas without needing to specify the exact parameters of resource competition, allowing us to infer the global properties of resource competition that can recapitulate experimentally observed time series statistics. We further show that our effective CR model captures the behavior of a broader class of ecological interactions, and can guide the development and analysis of other models and their time series statistics. Our work thus provides an accessible connection between complex microbiotas and the effective resource competition that could underlie their dynamics, with broad applications for engineering communities relevant to human health and to agriculture.

## Results
### A coarse-grained CR model under fluctuating environments

To determine the nature of time series statistics generated by resource competition, we considered a minimal CR model in which $N$ consumers compete for $M$ resources via growth dynamics described by

$$\frac{dX_i}{dt} = X_i \sum_{j=1}^{M} R_{ij} Y_j,$$

$$\frac{dY_j}{dt} = -Y_j \sum_{i=1}^{N} R_{ij} X_i. \tag{1}$$

Here, $X_i$ denotes the abundance of consumer $i$, $Y_j$ the amount of resource $j$, and $R_{ij}$ the consumption rate of resource $j$ by consumer $i$. The resources in this model are defined at a coarse-grained level, such that individual resources represent effective groups of metabolites or niches. We assumed that the resource consumption rates $R_{ij}$ were independent of the external environment and constant over time, thereby specifying the intrinsic ecological properties of the community with a collection of $N \times M$ microscopic parameters. To simplify this vast parameter space, we conjectured that the macro-ecological features of our experimental time series might be captured by typical profiles of resource consumption drawn from a statistical ensemble. This is a crucial simplification: while these randomly drawn values will never match the specific resource consumption rates of a given microbiota, previous work suggests that they can often recapitulate the large-scale behavior of sufficiently diverse communities (*Cui et al., 2021*). This simplification allows us to test whether particular ensembles of resource consumption rates can reproduce the time series statistics we observe. Specifically, we considered an ensemble in which each $R_{ij}$ was randomly selected from a uniform distribution between 0 and $R_{\max}$. To model the sparsity of resource competition within the community, each $R_{ij}$ was set to zero with probability $S$ (*Figure 1A*). This ensemble approach allows us to represent arbitrarily large communities with just two global parameters, $S$ and $R_{\max}$.

We simulated the dynamics in *Equation 1* using a serial dilution scheme (*Erez et al., 2020*) to mimic the punctuated turnover of gut microbiotas due to multiple feedings and defecations between sampling times. During a sampling interval $T$, each dilution cycle was seeded with an initial amount of each resource, $Y_{j,0}(T)$, and *Equation 1* was simulated until all resources were depleted ($dY_j/dt = 0$ for all $j$). The community was then diluted by a factor $D$ and resources were replenished to their initial amounts $Y_{j,0}(T)$ (*Figure 1B*). To mimic the effects of a reservoir of species that could potentially compete for the resources (*Ng et al., 2019*), we initialized the first dilution cycle of each sampling interval by assuming that $N$ consumers were present at equal abundance. Additional dilution cycles were then performed until an approximate ecological steady state was reached (*Figure 1B*, Materials and methods). Consumer abundances at sampling time $T$ were defined by this approximate ecological steady state. For the relevant parameter regimes we considered, this approximate steady state was reached within a reasonable number of generations (5–6 dilutions or ~40 generations for $D = 200$). Although the precise details of microbiota turnover are largely unknown in humans, our modeling results were robust to the precise value of $D$ and threshold for ecological steady state (*Figure 1—figure supplement 1*). Similarly, our results did not depend on the precise composition of the reservoir (*Figure 1—figure supplement 2*), although they did depend on its existence and relative size (*Figure 1—figure supplement 2*).

Under the assumptions of this model, any temporal variation in consumer abundances must arise through external fluctuations in the initial resource levels $Y_{j,0}(T)$, which might come, for example, from dietary fluctuations. To model these fluctuations, we assumed that the initial resource levels undergo a biased random walk around their average values $\bar{Y}_j$:

$$Y_{j,0}(T) = \left| Y_{j,0}(T-1) - k\left( Y_{j,0}(T-1) - \bar{Y}_j \right) + \sigma \bar{Y}_j \xi_j(T) \right|, \tag{2}$$

where $\xi_j(T)$ is a normally distributed random variable with zero mean and unit variance, $\sigma$ determines the magnitude of resource fluctuations, and $k$ is the strength of a restoring force that ensures the same resource environment on average over time (*Figure 1A*). The absolute value enforces $Y_{j,0}$ to be positive. If $k = 0$, there is no restoring force and hence $Y_{j,0}(T)$ performs an unbiased random walk; if $k = 1$, $Y_{j,0}(T)$ fluctuates about its set point $\bar{Y}_j$ independent of its value at the previous sampling time. For all $k > 0$, the model exhibits long-term stability without drift. As above, we used an ensemble approach to model the set points $\bar{Y}_j$, assuming that each $\bar{Y}_j$ was independently drawn from a uniform distribution between 0 and $Y_{\max}$. These assumptions yield a Markov chain of fluctuating resource

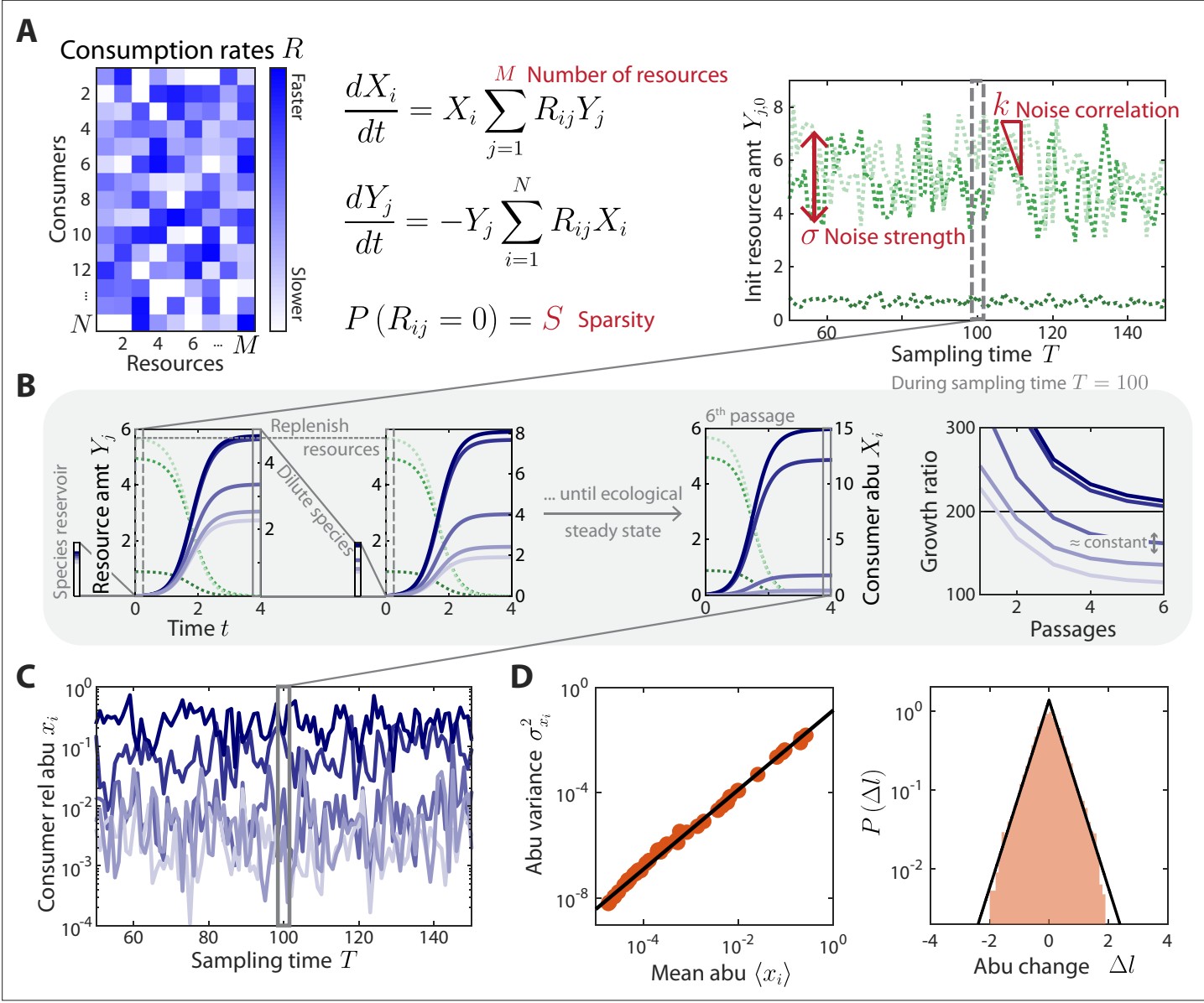

**Figure 1.** A coarse-grained consumer-resource model with fluctuating resource amounts. (**A**) In the consumer-resource model, $X_i$ denotes the abundance (abu) of consumer $i$ and $Y_j$ denotes the amount of coarse-grained resource $j$. The dynamics of the model are specified by consumption rates $R_{ij}$ for $N$ consumers and $M$ resources. $R_{ij}$ is drawn from a uniform distribution, and each $R_{ij}$ is set to zero with probability $S$, the sparsity of resource competition. The initial resource amount $Y_{j,0}(T)$ at each sampling time $T$ fluctuates with noise strength $\sigma$ and restoring force $k$. $N$ is estimated from each data set, and the four free ensemble level parameters are highlighted in red. (**B**) Shown are the dynamics of the model within one sampling time ($T = 100$, dashed gray box) for a subset of consumers and resources in a typical simulation. At each sampling time $T$, the model was simulated under a serial dilution scheme in which consumers (solid blue lines) grew until all resources (dotted green lines) were depleted, after which all consumer abundances were diluted by a fixed factor $D = 200$ and resource amounts were replenished to $Y_0(T)$. Each sampling time was initiated from an external reservoir of consumers, with all consumers present at equal abundance. Dilutions were repeated until an approximate ecological steady state was reached in which the ratios of final to initial abundances of all consumers changed by less than 5% of $D$ between subsequent dilutions (Materials and methods). The relative abundances at sampling time $T$ were obtained from the final species abundances at steady state. (**C**) The model maps a set of fluctuating resource amounts $Y_{j,0}(T)$ to a time series of consumer relative abundances $x_i(T)$ that can be compared to experimental measurements. (**D**) The simulated time series in (**C**) exhibits statistical behaviors that reproduce those found in experiments, including a power law scaling between the abundance variance and mean over time of each species (left) and an approximately exponential distribution of abundance changes (right). Black lines denote the best linear fit (left) and the best fit exponential distribution (right). The simulation shown in (**A–D**) was generated with $(N, M, S, \sigma, k) = (50, 30, 0.1, 0.2, 0.8)$.

The online version of this article includes the following figure supplement(s) for figure 1:

*Figure 1 continued on next page*

*Figure 1 continued*

**Figure supplement 1.** The dilution factor and steady-state threshold do not substantially affect time series statistics.

**Figure supplement 2.** Reservoir composition does not substantially affect time series statistics.

amounts $Y_{j,0}(T)$ and their corresponding consumer relative abundances $x_i(T) = X_i(T) / \sum_n X_n(T)$ (*Figure 1C*).

The statistical properties of these time series are primarily determined by five global parameters: the total number of consumers in the reservoir $N$, the number of resources in the environment $M$, the sparsity $S$ of the resource consumption matrix, and the resource fluctuation parameters $\sigma$ and $k$. The absolute magnitudes of $R_{max}$ and $Y_{max}$ are not important for our purposes since they do not affect the predictions of consumer relative abundances at ecological steady state. We extracted $N$ from experimental data as the number of consumers that were present for at least one sampling time point, leaving only four free global parameters.

Previous studies have suggested that the family level is an appropriate coarse graining of metabolic capabilities (*Goldford et al., 2018*; *Louca et al., 2016*; *Tian et al., 2020*), thus we assumed, unless otherwise specified, that each consumer grouping $i$ within our model represents a taxonomic family, and combined abundances of empirical operational taxonomic units (OTUs) or amplicon sequencing variants (ASVs; *Callahan et al., 2016*) at the family level for analyses (Materials and methods). Given the typical limits of detection of 16S amplicon sequencing data sets, we only examined time series statistics for taxa with relative abundance $>10^{-4}$ at any given time point. Experimental and simulated data were processed equivalently to enable consistent comparisons of their time series statistics.

As expected, we found that random realizations of our model (i.e., different resource consumption matrices drawn from the same ensemble) generated similar time series statistics, whose typical behavior strongly varied with the global parameters of the model. In particular, only small subsets of the parameters led to time series statistics that agreed with experiments, as we show below. An example simulation using the macroscopic parameters $(N, M, S, \sigma, k) = (50, 30, 0.1, 0.2, 0.8)$ is shown in *Figure 1*. This set of parameters produced relative abundance time series with highly similar statistical behaviors as in experiments involving daily sampling of human stool (*Figure 1D*). Given this agreement, we next systematically analyzed the time series statistics generated by our model across the macroscopic parameter space and compared against experimental behaviors to estimate model parameters for wide-ranging microbiotas.

## Model reproduces the statistics of human gut microbiota time series

To test whether our model can recapitulate major features of experimental time series, we first focused on a data set of daily sampling of the gut microbiota from a human subject (*Caporaso et al., 2011*; *Figure 2*). These data were previously shown (*Ji et al., 2020*) to exhibit several distinctive statistical behaviors: (1) the variance $\sigma_{x_i}^2$ of family $i$ over the sampling period scaled as a power law with its mean $\langle x_i \rangle$ (*Figure 2B and F*); (2) the $\log_{10}$(abundance change) $\Delta l_i(T) = \log_{10}(x_i(T+1)/x_i(T))$, pooled over all families and across all sampling times, was well fit by an exponential distribution with standard deviation $\sigma_{\Delta l}$ (*Figure 2B and G*); and (3) the distributions of residence times $t_{res}$ and return times $t_{ret}$ (the durations of sustained presence and absence, respectively) pooled over all families were well fit by power laws with an exponential cutoff (*Figure 2D and K*). Through an exhaustive search of parameter space, we identified a specific combination of parameters that could reproduce all of these behaviors within our simple CR model (*Figure 2F, G and K*).

In addition, several other important statistics were reproduced without any additional fitting: (1) the distribution of richness $\alpha(T)$, the number of consumers present at sampling time $T$ (*Figure 2A and E*); (2) the distribution of the restoring slopes $s_i$ of the linear regression of $\Delta l_i(T)$ against $l_i(T) \equiv \log_{10}(x_i(T))$ across all $T$ (*Figure 2C and H*); (3) the distribution of prevalences $p_i$, the fraction of sampling times for which family $i$ is present (*Figure 2A1*); (4) the relationship between $p_i$ and $\langle x_i \rangle$ (*Figure 2J*); and (5) the rank distribution of mean abundances $\langle x_i \rangle$ (*Figure 2L*).

Therefore, our model was able to simultaneously capture at least eight statistical behaviors in a microbiota time series with only four parameters, each of which may represent biologically relevant features of the community.

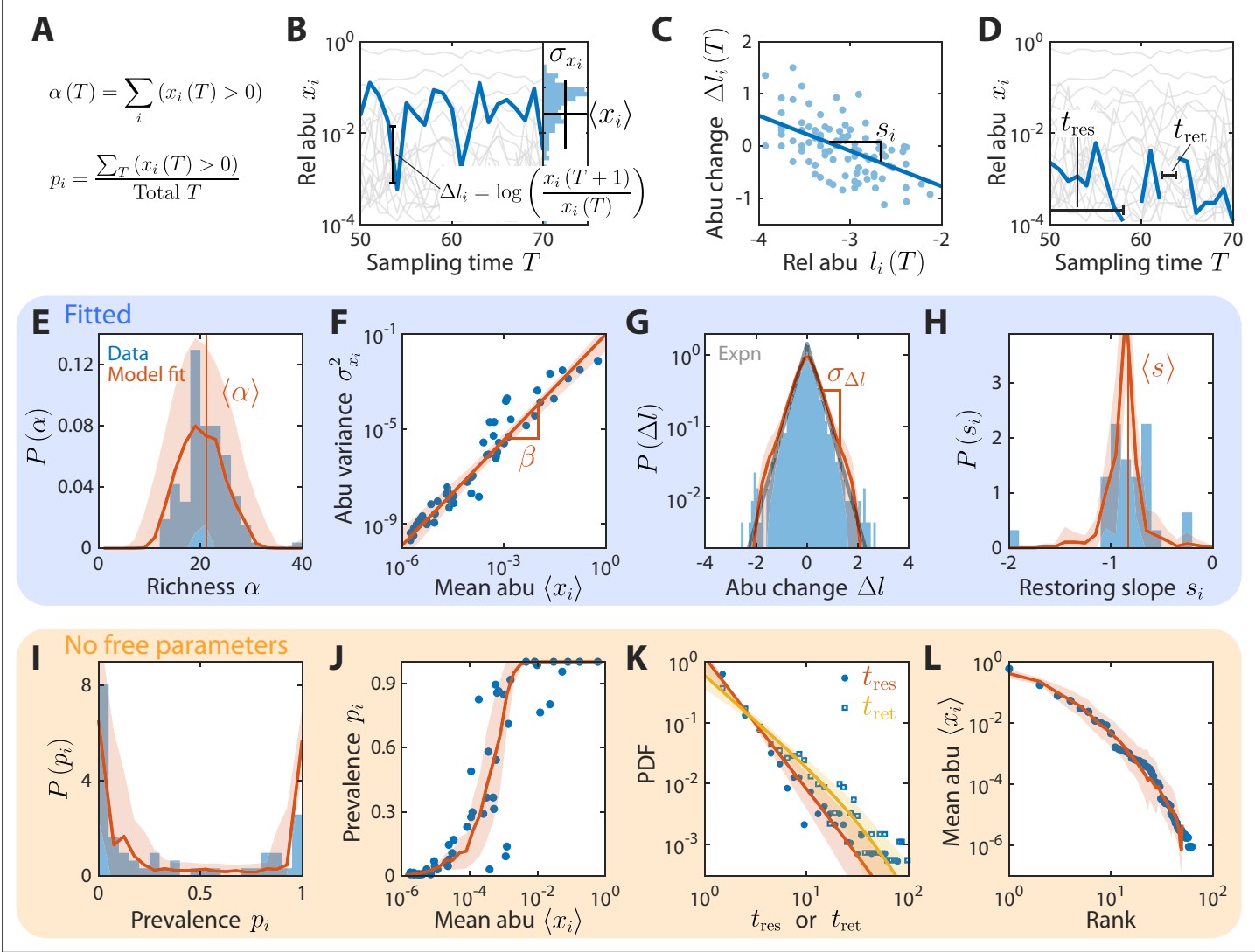

**Figure 2.** A coarse-grained consumer-resource model with fluctuating resource amounts reproduces experimentally observed statistics in an abundance time series from daily sampling of a human gut microbiota. In all panels, blue points and bars denote experimental data analyzed and aggregated at the family level (*Caporaso et al., 2011*). Red lines and shading denote best fit model predictions as the mean and standard deviation, respectively, across 20 random instances of the best fit ensemble level parameters, $(N, M, S, \sigma, k) = (50, 30, 0.1, 0.2, 0.8)$. (**A–D**) Illustrations of various time series statistics in (**E–L**). (**A**) The distribution of richness $\alpha$, the number of consumers present at a sampling time, and its mean $\langle\alpha\rangle$ are well fit by the model. (**B**) The variance $\sigma_{x_i}^2$ and mean $\langle x_i \rangle$ over time of each family's abundance (abu) scale as a power law with exponent $\beta$. Here, $\beta = 1.48$ in experimental data and in simulations. (**C**) The distribution of $\log_{10}$(abundance change) $\Delta l$ across all families is well fit by an exponential with standard deviation $\sigma_{\Delta l}$. The gray line denotes the best fit exponential distribution, and is largely overlapping with the model prediction in red. (**D**) The distribution of restoring slopes $s_i$, defined based on the linear regression between the abundance change and the relative abundance for a species across time, is tightly distributed around a mean $\langle s \rangle$ that reflects the environmental restoring force. Best fit values of model parameters were determined by minimizing errors in $\langle\alpha\rangle$, $\beta$, $\sigma_{\Delta l}$, and $\langle s \rangle$ (E–H, respectively). Using these values, our model also reproduced the distribution of prevalences (fraction of sampling times in which a consumer is present, **I**), the relationship between prevalence and mean abundance (**J**), the distributions of residence and return times (durations of sustained presence or absence, respectively, as illustrated in **D**) (**K**), and the rank distribution of abundances (**L**).

The online version of this article includes the following figure supplement(s) for figure 2:

**Figure supplement 1.** Grouping at a coarser taxonomic level results in similar time series statistics.

**Figure supplement 2.** The consumer-resource (CR) model can reproduce time series statistics at the genus level of a human gut microbiota.

To determine whether our model can be used to analyze time series statistics at other taxonomic levels, we analyzed the same data set (*Caporaso et al., 2011*) at finer (genus) and coarser (class) taxonomic levels, both of which exhibited qualitatively similar statistical behaviors as the family level. Our modeling framework was able to quantitatively recapitulate almost all statistics at both levels

(*Figure 2—figure supplements 1 and 2*). A notable exception is that the *Bacteroides* genus dominated the observed rank abundance distribution at the genus level, while our CR model predicted a more even distribution (*Figure 2—figure supplement 2*). Nevertheless, the relative abundances among the remaining genera were still well captured by the model predictions (*Figure 2—figure supplement 2*). These results demonstrate that our model and its applications can be generalized across taxonomic levels.

## Systematic characterization of the effects of CR dynamics on time series statistics

Since our model can reproduce the observed statistics in gut microbiota time series, we sought to determine how these statistics would respond to changes in model parameters, and thus how experimental measurements constrain the ensemble parameters across various data sets. To do so, we simulated our model across all relevant regions of parameter space. $S$ and $k$ were varied across their entire ranges, and $M$ and $\sigma$ were varied across relevant regions outside of which the model clearly disagreed with the observed data. For each set of parameters, each time series statistic was averaged across random instances of $R_{ij}$ and $Y_{j,0}(T)$ drawn from the same statistical ensemble. For each statistic $z$, its global susceptibility $C(z, w)$ to parameter $w$ was calculated as the change in $z$ when $w$ is varied, averaged over all other parameters and normalized by the standard deviation of $z$ across the entire parameter space. Due to the normalization, $C(z, w)$ varies approximately between –3 and 3, where a magnitude close to 3 indicates that almost all the variance of $z$ is due to changing $w$.

By clustering and ranking susceptibilities, we identified four statistics with $|C(z, w)| > 2$ that were largely determined by one of each of the four model parameters (*Figure 3*, *Figure 3—figure supplement 1*): mean richness $\langle \alpha \rangle$, the power law exponent $\beta$ of $\sigma_{x_i}^2$ versus $\langle x_i \rangle$, the standard deviation in $\log_{10}$(abundance change) $\sigma_{\Delta l}$, and the mean restoring slope $\langle s \rangle$ were almost exclusively susceptible to variations in $M$, $S$, $\sigma$, and $k$, respectively. Similar results were also obtained for local versions of the susceptibility, in which individual parameters were varied around the best fit values for the human gut microbiota in *Figure 2* (*Figure 3-figure supplement 2*). These susceptibilities broadly illustrate how various time series statistics are affected by coarse-grained parameters of resource competition; we further investigate some specific examples in the next section.

The exclusive susceptibilities of these four statistics suggest that they can serve as informative metrics for estimating model parameters. Therefore, we estimated model parameters by minimizing the sum of errors between model predictions and experimental measurements of these four statistics, and obtained estimation bounds by determining parameter variations that would increase model error by 5% of the mean error across all parameter space. As we will show, the resulting bounds are small relative to the differences among distinct microbiotas, indicating that meaningful conclusions can be drawn from the best fit values of the ensemble level parameters of resource competition. In summary, the four model parameters were fit to four summary statistics: mean richness $\langle \alpha \rangle$, variance-mean scaling exponent $\beta$, standard deviation of abundance change $\sigma_{\Delta l}$, and mean restoring slope $\langle s \rangle$ (*Figure 2E–H*, respectively). The shapes of their corresponding distributions and scalings, as well as at least four other statistics (*Figure 2I–L*), are all parameter-free predictions.

## Origins of distinctive statistical behaviors in species abundance time series

To understand the mechanisms that underlie the susceptibilities of various time series statistics to model parameters, we investigated their origins within our model, focusing on how they constrain the parameters.

The average richness $\langle \alpha \rangle$ is a fundamental descriptor of community diversity. Within our model, $\langle \alpha \rangle$ is largely determined by and increases with increasing resource number $M$ ($C(\alpha, N/M) = -2.6$), as expected for CR dynamics. The sparsity of resource use $S$ impacts the power law exponent $\beta$ between $\sigma_{x_i}^2$ and $\langle x_i \rangle$ ($C(\beta, S) = -2.0$). Together, $\alpha$ and $\beta$ constrain the parameters of resource competition $M$ and $S$.

The effect of $S$ on $\beta$ can be partially understood by considering limiting behaviors as follows. When sparsity is high ($S \approx 1$), there is little competition and each consumer consumes almost distinct sets of resources from other consumers. In the limit in which each consumer utilizes a single unique resource, $\sigma_{x_i}^2$ is determined by the noise in resource level, which has a $\beta = 2$ scaling according to *Equation 2*.

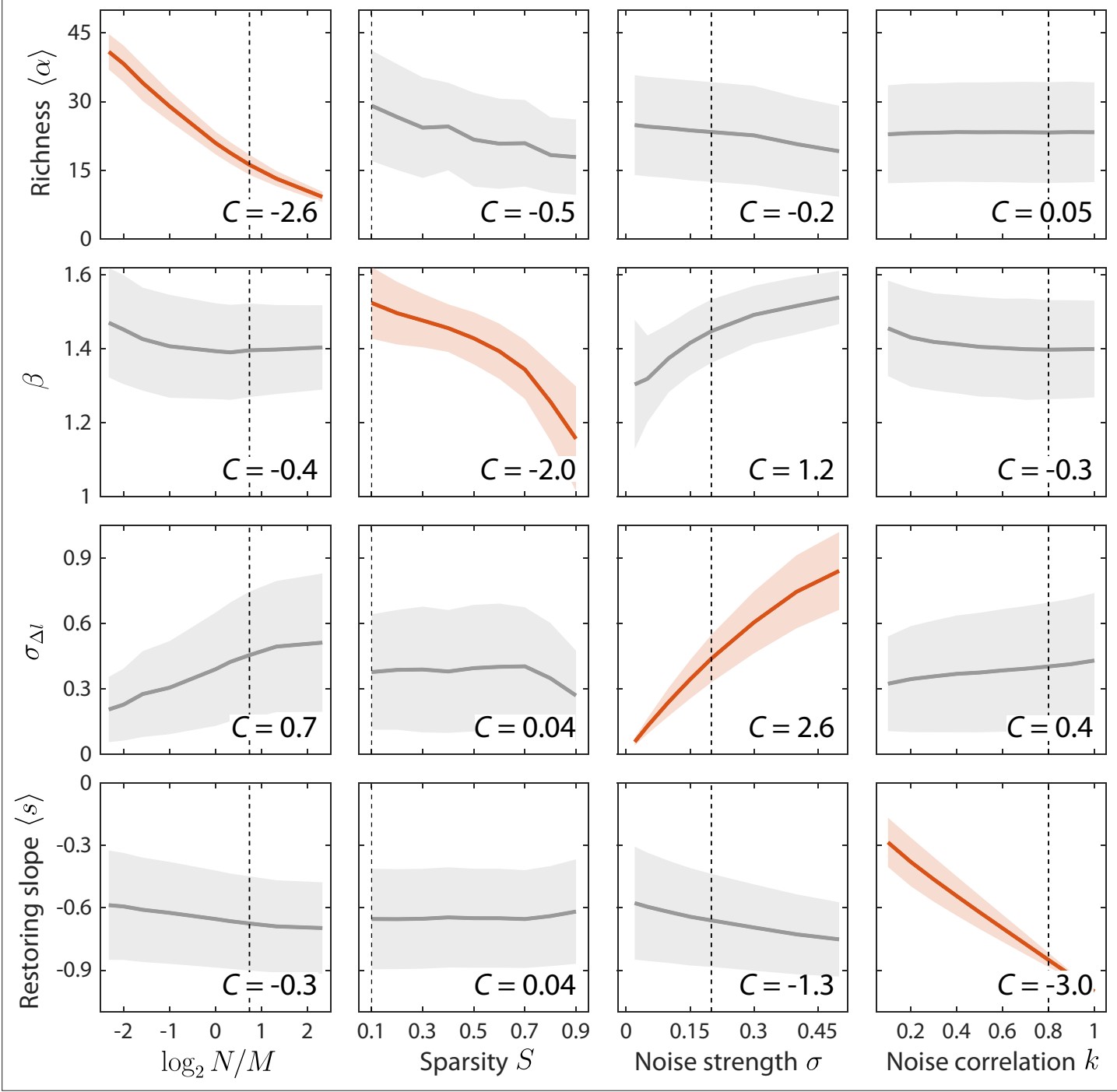

**Figure 3.** Macroscopic parameters of resource competition affect time series statistics in distinct manners. Shown are the changes in time series statistics (y-axis) in response to changes in model parameters (x-axis) for a comprehensive search across relevant regions of parameter space. Lines and shading show the mean and standard deviation of a statistic at the given parameter value across variations in all other parameters. Data are plotted in red when the corresponding susceptibility $|C(z, w)| > 2$, indicating that statistic $z$ is strongly affected by parameter $w$ regardless of the values of other parameters. Dashed lines highlight best fit parameter values to the experimental data in *Figure 2*. Simulations were carried out for $N = 50$ across $M \in [10, 20, 30, 40, 50, 100, 150, 200, 250]$, $S \in [0.1, 0.9]$ in 0.1 increments, $\sigma \in [0.05, 0.5]$ in 0.05 increments, and $k \in [0.1, 1]$ in 0.1 increments.

The online version of this article includes the following figure supplement(s) for figure 3:

**Figure supplement 1.** Time series statistics are differentially susceptible to model parameters.

**Figure supplement 2.** Local susceptibilities behave similarly to their global counterparts.

*Figure 3 continued on next page*

*Figure 3 continued*

**Figure supplement 3.** A no-competition model provides a partial explanation for the scaling exponent $\beta$ between $\sigma_{x_i}^2$ and $\langle x_i \rangle$ .

**Figure supplement 4.** Sparsity and the number of metabolites determine the shape of the distribution of abundance changes $P\left(\Delta l\right)$ .

**Figure supplement 5.** Our consumer-resource (CR) model is consistent with results obtained by shuffling time labels.

In the limit of large $M$ and high sparsity, the variation in the number of resources consumed by each consumer can be large relative to the mean, and both $\sigma_{x_i}^2$ and $\langle x_i \rangle$ scale with the number of resources consumed, hence $\beta = 1$. Simulations of a no-competition model in which consumers consume distinct sets of resources confirmed the scalings in these limits (*Figure 3—figure supplement 3*). By contrast, when sparsity is low ($S \approx 0$), each consumer utilizes almost all resources and hence variation in the number of resources consumed is small relative to the mean. Despite the obvious presence of competition in our CR model, we nevertheless attempted to understand the low sparsity limit by extrapolating the no-competition model above to a case in which all consumers consume distinct sets of the same number of resources. For large number of resources, these simulations predicted that $\beta \approx 1.5$ (*Figure 3—figure supplement 3*), as did our CR model for $S = 0.1$ (*Figure 3—figure supplement 3*). These findings suggest that the effect of $S$ on $\beta$ can be partially attributed to differences in the number of resources consumed.

The distribution of $\Delta l$ describes the nature of abundance changes. As expected, the width of the distribution is largely determined by and increases with increasing $\sigma$ ($C\left(\sigma_{\Delta l}, \sigma\right) = 2.6$). For the gut microbiota data set in *Figure 2*, the shape of the distribution was well fit by an exponential. Within our model, the shape of the distribution aggregated across all consumers is determined by $N/M$ and the sparsity $S$, emerging from the mixture of each consumer's individual distribution (*Figure 3—figure supplement 4*). When $N/M < 1$ and the sparsity $S$ is low, individual distributions of $\Delta l$ are well fit by normal distributions, and pool together to generate another normal distribution. When $N/M < 1$ and sparsity $S$ is high, individual distributions remain normal, but can pool together to generate a non-normal distribution that is well fit by an exponential (see also *Allen et al., 2001*). By contrast, when $N/M > 1$, individual distributions can be well fit by an exponential and can pool together to approximate another exponential. Simulations of the no-competition model considered above led to individual and aggregate distributions that were normal in all cases, indicating that in our model resource competition is responsible for generating the non-normal distributions of $\Delta l$ (*Figure 3—figure supplement 3*). Although it is challenging to discern the shape of individual distributions in most experimental data sets given the limited numbers of samples, the shape of the aggregate distribution of $\Delta l$ informs the parameters of resource competition $M$ and $S$. In particular, an exponential distribution of $\Delta l$ suggests either strong resource competition in the form of $N > M$ or substantial niche differentiation in the form of high $S$. Other statistics such as $\beta$ can help to distinguish between these two regimes.

The distribution of restoring slopes $s_i$ describes the tendency with which consumers revert to their mean abundances following fluctuations. As expected, the mean $\langle s \rangle$ is almost completely determined by $k$, which describes the autocorrelation in resource levels ($-\langle s \rangle \approx k$ and $C\left(\langle s \rangle, k\right) = -3.0$). Together, the distributions of $\Delta l$ and $s_i$ constrain the parameters of external fluctuations $\sigma$ and $k$.

Within our model, resource fluctuations can lead to the temporary 'extinction' of certain species when they drop below the detectability threshold of $10^{-4}$. The distributions of residence and return times, $t_{\mathrm{res}}$ and $t_{\mathrm{ret}}$, reflect the probabilities of extinction as well as correlations between sampling times. For all parameter sets explored, these distributions can be well fit by power laws, with an exponential cutoff to account for finite sampling (*Ji et al., 2020*). As expected, the power law slopes $\nu_{\mathrm{res}}$ and $\nu_{\mathrm{ret}}$ decrease (become more negative) with increasing $\sigma$ or $k$ (*Figure 3—figure supplement 1*), since increasing external noise or decreasing correlations in time increases the probability of fluctuating between existence and extinction for each consumer. By contrast, $\nu_{\mathrm{res}}$ and $\nu_{\mathrm{ret}}$ change in opposite directions in response to variation in $M$ (*Figure 3—figure supplement 1*). Increasing $M$ leads to a larger number of highly prevalent consumers, thereby increasing the mean and broadening the distribution of $t_{\mathrm{res}}$ and decreasing the mean and narrowing the distribution of $t_{\mathrm{ret}}$. Since the four ensemble level parameters are already fixed by other statistics, the distributions of $t_{\mathrm{res}}$, $t_{\mathrm{ret}}$, and $p_i$ are parameter-free predictions of our model. In other words, a macroscopic characterization of the effective resource competition and resource fluctuations is sufficient to predict the statistics of 'extinction' dynamics, as

well as the abundance rank distribution and the relationship between consumer abundance and prevalence.

Since the distributions of $\Delta l$, $t_{\text{res}}$, and $t_{\text{ret}}$ are dependent on correlations between sampling times, it was initially puzzling that their distributions in some data sets remained similar after shuffling sampling times, raising questions as to what extent these statistics hold information about the underlying intrinsic dynamics (*Tchourine et al., 2021*; *Wang and Liu, 2021a*). Our results assist in reconciling the apparent conundrum, since within our model richness $\langle \alpha \rangle$ and Taylor's law exponent $\beta$ do not depend on correlations between sampling times and are also the statistics that are most informative about the intrinsic parameters $M$ and $S$ (*Figure 3*). As a result, the shuffled time series were also well fit by our model and yielded best fit values that were identical to those produced by the actual time series except with $k = 1$, as expected due to the absence of correlation across sampling times (*Figure 3—figure supplement 5*). Thus, our results suggest that while external fluctuations in resource levels may be responsible for generating species abundance variations, the intrinsic properties of resource competition can determine the resulting scaling exponents of many statistical behaviors.

Taken together, our analyses demonstrate the complex relationships among time series statistics and highlight their unification within our model using only a small number of global parameters, whose values are strongly constrained by macro-ecological patterns.

## CR model guides the identification of other models that can reproduce time series statistics

We have shown that many time series statistics can be recapitulated by a simple model that does not require knowing many detailed features of real microbiota (*Figure 2*). The success of this approach implies that these macroecological fluctuations must be independent of at least some model details, which suggests that there may be other ecological models that could also recapitulate the same data (*Figure 4*, *Figure 4—figure supplements 1–5*). The relationships between ecological models are generally poorly characterized. To explore these possibilities, we sought to compare our calibrated CR models against several common alternatives.

First, we aimed to determine the extent to which the simulated statistics depend on the assumptions of our CR model. Our parametrization of the consumption rates introduces a correlation between the maximum growth rate of a consumer and

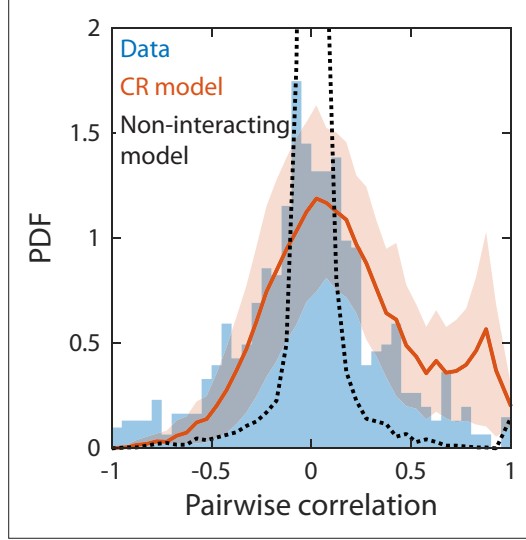

**Figure 4.** Correlations between abundances of consumer pairs were captured by the consumer-resource model, but not by a null model without interspecies interactions. Shown in blue is the probability density function (PDF) of correlations between the abundances across sampling times of all consumer pairs for the experimental data in *Figure 2*. Red line represents parameter-free model predictions as in *Figure 2*, using the same best fit parameters; shading represents 1 standard deviation. Black dashed line shows predictions of a null model without interspecies interactions in which consumer abundances were drawn from independent normal distributions whose mean and variance were extracted from data.

The online version of this article includes the following figure supplement(s) for figure 4:

**Figure supplement 1.** The consumer-resource (CR) model with metabolic trade-offs produces similar statistics as the original model and can also recapitulate experimental data.

**Figure supplement 2.** Consumer-resource (CR) model with saturation kinetics exhibits dampened fluctuations but can still reproduce experimentally observed statistics.

**Figure supplement 3.** A non-interacting null model reproduced some, but not all, time series statistics.

**Figure supplement 4.** Generalized Lotka-Volterra (gLV) model with consumer-resource (CR)-converted interaction coefficients generates time series statistics similar to the original CR model and also recapitulates experimental data.

**Figure supplement 5.** Generalized Lotka-Volterra (gLV) model with normally distributed interaction coefficients cannot reproduce experimental data.

the number of resources it consumes. To remove this correlation, we normalized the sum of consumption rates $\sum_j R_{ij}$ for consumer $i$ to a fixed capacity $\widetilde{R}_i$ that was randomly drawn from the original growth rates $\sum_j R_{ij} Y_{j,0}$ (*Good et al., 2018*; *Posfai et al., 2017*; *Tikhonov and Monasson, 2017*). This modification preserves the variation in consumer fitness while implementing a metabolic trade-off. The resulting time series statistics were essentially unaffected, also recapitulating experimental data (*Figure 4—figure supplement 1*).

Moreover, the CR dynamics in *Equation 1* do not consider other biologically plausible scenarios such as saturation kinetics (*Momeni et al., 2017*; *Niehaus et al., 2019*). To probe the robustness of the results of our model to the dynamical assumptions, we implemented saturation kinetics with all other details kept the same (Materials and methods). When this model was simulated with the best fit parameters of the original model, the resulting dynamics were less variable across sampling times than without saturation kinetics, since the saturated regime is unaffected by small changes in resource levels (*Figure 4—figure supplement 2*). Nonetheless, experimental statistics were again reproduced once the strength of environmental fluctuations $\sigma$ was increased appropriately (*Figure 4—figure supplement 2*). This suggests that our results are robust to assumptions regarding metabolic trade-offs and saturation kinetics.

We next considered a non-interacting null model in which consumer abundances were drawn from independent normal distributions whose means and variances were fitted directly from the data. Even with a large number of free parameters, this null model was unable to capture some of the time series statistics reproduced by our CR model, including Taylor's law as well as the distributions of richness and restoring slopes (*Figure 4—figure supplement 3*). We reasoned that the discrepancies between experimental data and the null model could be due to the lack of interspecies interactions. To test this hypothesis, we examined the pairwise correlations between the consumer abundances across sampling times. The measured distribution of pairwise correlations is much broader than the prediction of the non-interacting model, which is sharply peaked about zero as expected (*Figure 4*). By contrast, the distribution of correlations predicted by our CR model without any additional fitting was in much closer agreement with the experimental data (*Figure 4*). These findings imply that interspecies interactions are required to capture important details of community dynamics.

While our CR model assumes pairwise interactions between consumers and resources, the effective interactions between consumers are not necessarily pairwise. To explore whether these higher-order contributions are necessary for recapitulating the data, we considered models explicitly based on pairwise interspecies interactions, which despite differences compared with CR models (*Momeni et al., 2017*) can also reproduce some properties of experimental time series (*Descheemaeker and de Buyl, 2020*; *Wang and Liu, 2021b*). To further explore the properties of models focused on pairwise interactions, we investigated gLV models in which $N$ taxa grow and interact via

$$\frac{dX_i}{dt} = X_i \left( r_i + \sum_{j=1}^{N} A_{ij} X_j - \Gamma(t) \right), \tag{3}$$

where $X_i$ denotes the relative abundance of taxon $i$, $r_i$ its growth rate, and $A_{ij}$ its interaction coefficient with taxon $j$. $\Gamma(t) = \sum_i r_i X_i + \sum_{i,j} A_{ij} X_i X_j$ is a normalizing term that ensures that the relative abundances always sum to one (*Joseph et al., 2020*). Since this classical model is generally unstable for randomly drawn interaction coefficients (*May, 1972*), we sought to focus on particular instances of the gLV model that were closest to our original CR model. This conversion between models was achieved by converting the consumption rates $R_{ij}$ and resource levels $Y_{j,0}$ at each sampling time $T$ to the growth rates $r_i$ and interaction coefficients $A_{ij}$ that characterize the dynamics when consumption rates are similar to the mean value (Materials and methods). This conversion results in negative, symmetric $A_{ij}$ whose magnitudes depend on the niche overlap between the interacting taxa (*Good et al., 2018*). Moreover, fluctuations in $Y_{j,0}$ result in corresponding fluctuations in both $r_i$ and $A_{ij}$ across $T$. These CR-converted gLV models generated time series statistics that reproduced the experimental data to a similar extent as the original CR model (*Figure 4—figure supplement 4*). In light of this correspondence, we asked whether more general ensembles of pairwise interaction could also reproduce the experimental data. We randomly selected $r_i$ and $A_{ij}$ values from normal distributions with means and variances equal to those in the CR-converted gLV models while enforcing symmetric and negative interactions. The resulting gLV models yielded a

poor fit to the data (*Figure 4—figure supplement 5*). Together, these results suggest that while pairwise interactions between taxa are likely sufficient to recapitulate the experimental data, their parameters must be drawn from particular ensembles that can be more simply described in the CR framework.

These examples reinforce that only a particular subset of models can recapitulate the data, and therefore, that the underlying community properties are highly constrained by macroecological dynamics. Moreover, our calibrated CR model can guide the parametrization of other models that can satisfy those constraints, while also identifying model features that are necessary for recapitulating data.

## Time series statistics distinguish wide-ranging microbiotas

Having developed a simple method to estimate parameters of our CR model that recapitulate time series statistics, we applied this method to data sets involving wide-ranging microbial communities. Although the various communities considered are drastically different in many aspects, we hypothesized that our CR model framework could still be applied to identify the statistical ensembles that can describe their macroecological dynamics. In addition to microbiotas from the human and mouse gut (*Caporaso et al., 2011*; *Carmody et al., 2015*; *David et al., 2014*), we examined communities from the human vagina (*Song et al., 2020*), human saliva (*David et al., 2014*), and in and around rice roots (*Edwards et al., 2018*). The time series statistics of these microbiotas varied broadly (*Figure 5A*). Nevertheless, our model successfully reproduced the experimental statistics across all communities (*Figure 5—figure supplements 1–6*), suggesting that simple CR models can capture many of the macroscopic features of these microbiotas.

The best fit parameters suggest that the effective resource competition dynamics occur in distinct regimes across microbiotas (*Figure 5B*). Human gut microbiotas were best described by $N > M$, suggesting that there are more species in the reservoir than resources in the environment, by contrast to mouse gut microbiotas that were best described by $N < M$. In terms of resource niche overlaps, human gut microbiotas were best fit with sparsity $S < 0.3$, while mouse gut microbiotas were best fit with $S > 0.3$, suggesting that on average, pairs of bacterial families are more metabolically distinct in the mouse versus the human gut.

Unlike gut microbiotas, a human saliva microbiota yielded best fit parameters $N \approx M$ and $S \approx 0.8$, suggesting that this community has access to abundant resources and that each effective resource is competed for by a small fraction of the extant bacterial families. All vaginal microbiotas were best fit with $S < 0.1$, suggesting intense resource competition.

Like vaginal microbiotas, microbial communities residing in the bulk soil around rice roots and in the associated rhizoplane and rhizosphere were well described by $S < 0.1$. By contrast, the community in the associated endosphere was best described by $S \approx 0.6$, suggesting that resource competition is less fierce within plant roots than around them.

In addition, inferences about the nature of environmental fluctuations can be made from the best fit values of $\sigma$ and $k$ (*Figure 5B*). Apart from the two vaginal microbiota data sets, the best fit values of $\sigma$ ranged from 0.1 to 0.3, indicating that changes in resource levels smaller than this magnitude will generate abundance changes that look like typical fluctuations. The best fit values of $k$ varied between 0.5 and 1 across data sets, suggesting that the dynamics of microbial communities occur faster than or comparable to the typical sampling frequency of longitudinal studies. While it is unclear whether the internal time scales are faster than the sampling frequency for all of these communities, simulation results were robust to the dilution factor and threshold change defining ecological steady state (*Figure 1—figure supplement 1*), two main factors that affect the relationship between the internal and sampling time scales.

Inferences about intrinsic parameters of resource competition and external parameters of environmental fluctuations were also consistent with expectations for in vitro passaging of complex communities derived from humanized mice (*Aranda-Díaz et al., 2022*). The resulting time series statistics were best fit by the smallest value of $\sigma$ among the data sets studied, indicating that the in vitro environment has relatively low noise across sampling times (as expected); the nonzero $\sigma$ presumably arises from technical variations that result in effective noise in resource levels. The best fit value of $M$ was larger than the reservoir size $N$, suggesting that there are many distinct resources in the complex medium used for passaging and consistent with the ability of more diverse inocula to support more diverse in

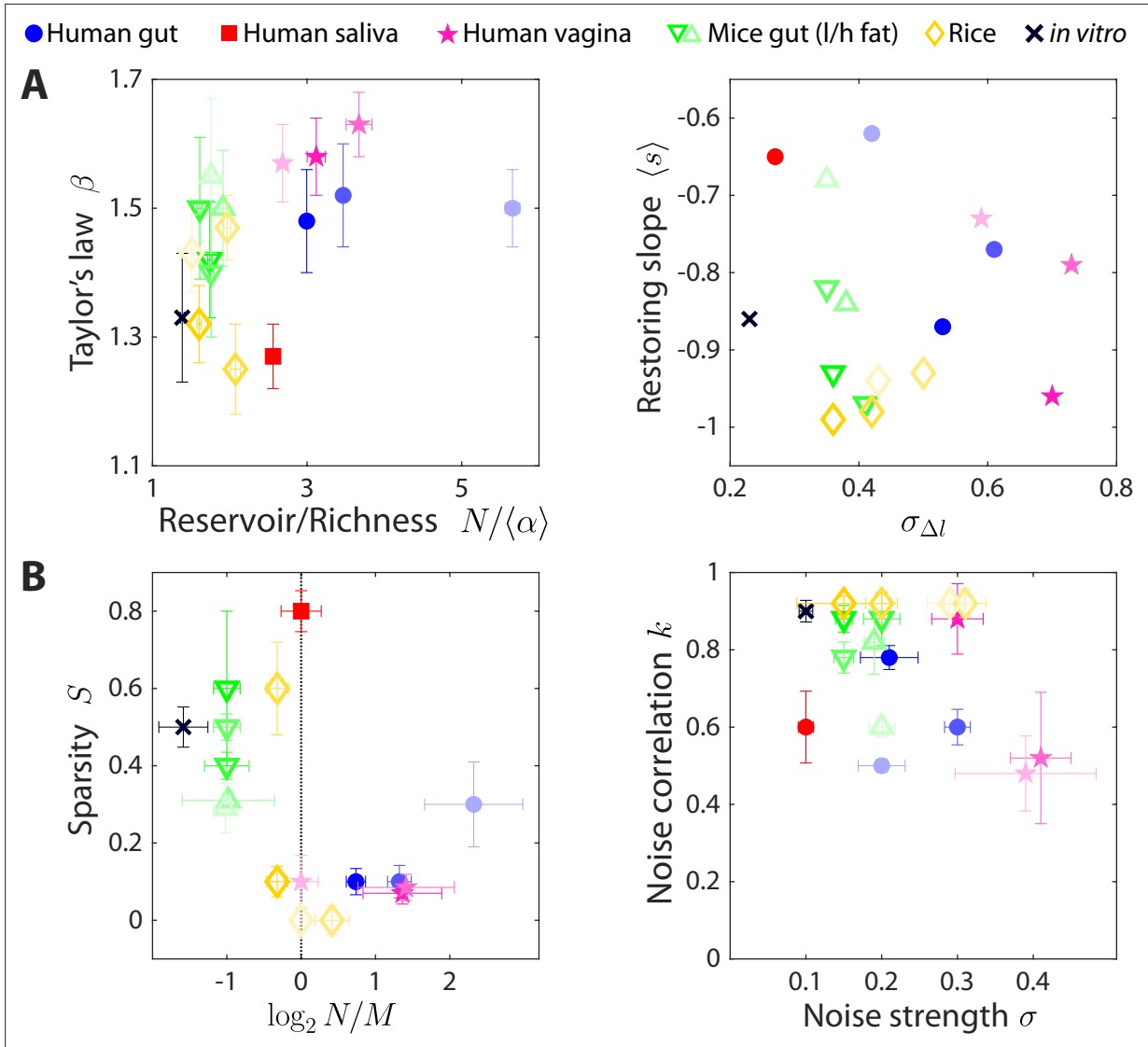

**Figure 5.** The statistics of wide-ranging microbiotas were captured by the coarse-grained consumer-resource model in different regimes of resource competition and environmental fluctuations. Shown are time series statistics (**A**) and corresponding best fit model parameters (**B**) for human microbiotas from stool (*Caporaso et al., 2011*; *David et al., 2014*) (blue circles), saliva (*David et al., 2014*) (red square), and the vagina (*Song et al., 2020*) (pink stars), gut microbiotas of mice under low fat (green downward triangles) and high fat (green upward triangles) diets (*Carmody et al., 2015*), and plant microbiotas from the rice endosphere, rhizosphere, rhizoplane, and bulk soil (*Edwards et al., 2018*) (diamonds). (**A**) Microbiota origin generally dictates the scaling exponent $\beta$ and the ratio between the reservoir size $N$ (number of observed families throughout the time series) and the richness $\langle \alpha \rangle$ (left), as well as the mean restoring slope $\langle s \rangle$ and standard deviation of $\log_{10}$(abundance change) (right). Error bars denote 95% confidence intervals. (**B**) Microbiota origin generally dictates the best fit parameters of resource competition, $N/M$ and $S$ (left), and of environmental fluctuations, $\sigma$ and $k$ (right). Error bars denote variation in the parameter that would increase model error (as interpolated between parameter values scanned) by 5% of the mean error across all parameter values scanned.

The online version of this article includes the following figure supplement(s) for figure 5:

**Figure supplement 1.** Model reproduces experimentally observed time series statistics in human gut microbiotas.

**Figure supplement 2.** Model reproduces experimentally observed time series statistics in a human saliva microbiota.

**Figure supplement 3.** Model reproduces experimentally observed time series statistics in human vagina microbiotas.

**Figure supplement 4.** Model reproduces experimentally observed time series statistics in mice gut microbiotas.

**Figure supplement 5.** Model reproduces experimentally observed time series statistics in rice microbiotas.

**Figure supplement 6.** odel reproduces experimentally observed time series statistics in an in vitro-passaged complex community.

vitro communities (*Aranda-Díaz et al., 2022*). The consistency of these results further supports the utility of our model.

Taken together, our model infers ensemble-level parameters of resource competition and external parameters of environmental fluctuations for several widely studied microbial communities that can inform future mechanistic studies.

## Discussion

Here, we presented a coarse-grained CR model that generates species abundance time series from fluctuating environmental resources. We demonstrated that this model reproduces several statistical behaviors (*Figure 2*) and elucidated how these observations constrain the parameters of resource competition within the model (*Figure 3*). Moreover, we successfully fitted the model to wide-ranging microbiotas, which allowed us to draw inferences about the parameters of their effective resource competition. In sum, our work provides an existence proof that a CR model can recapitulate experimentally observed time series statistics in microbiotas from diverse environments.

An important feature of our model is that it does not need to specify the individual resource uptake rates of different taxa, which could be too numerous and complex to be tractable. Instead, our model reproduces many statistical behaviors with a small number of global parameters that describe the distributions of resource uptake rates. To what extent these macroscopic parameters can be interpreted mechanistically is an interesting open question that could be explored in future work. Although by no means exhaustive, our framework nevertheless addresses several pertinent questions regarding construction of useful models of microbiota dynamics. The success of our CR model in reproducing experimental time series statistics is consistent with bioinformatics-guided analyses of complex communities demonstrating that metabolic capability is a major determinant of community composition (*Louca et al., 2016*; *Tian et al., 2020*). Our results also suggest that the contributions of a reservoir of species or other forms of species re-introduction are important for the dynamics of wide-ranging microbiotas. Within our model, the lack of species re-introduction renders poor consumers unable to recover to meaningful abundance within a sampling time even when resource fluctuations are in their favor, thereby distorting time series statistics. The existence of a reservoir is consistent with previous experimental work in mice (*Ng et al., 2019*), but further work is required to investigate how species re-introduction occurs in other systems. Similarly, further experimental work is required to ascertain the amount of growth and change that occurs during sampling time scales, and further theoretical work is required to infer such internal time scales from microbiota time series.

In terms of intrinsic metabolic properties, our results provide a baseline expectation for the effective number of resources or available niches in the wide-ranging systems examined here, and to what extent they are competed for by extant consumers. In terms of environmental properties, our results provide a baseline expectation to help distinguish between typical fluctuations and large perturbations in resources. These expectations may aid in the engineering of complex microbiotas.

In general, our work demonstrates that it is feasible to reproduce time series statistics using CR models of microbiota dynamics, thereby generating mechanistic hypotheses for further investigation. Our CR model and fitting procedure can also be used to aid the parametrization of other models such as Lotka-Volterra models (*Figure 4—figure supplements 1–5*), comparisons among which can reveal the model details that are required to recapitulate experimental data. In the future, more detailed hypotheses can be generated by investigating how time series statistics are affected by modifications to baseline CR dynamics, such as the incorporation of metabolic cross-feeding (*Goldford et al., 2018*; *Li et al., 2020*) or physical interactions such as type VI killing (*Verster et al., 2017*), functional differentiation from genomic analysis (*Arkin et al., 2018*; *Machado et al., 2021*; *Pollak et al., 2021*), and physical variables such as pH (*Aranda-Díaz et al., 2020*; *Ratzke and Gore, 2018*), temperature (*Lax et al., 2020*), and osmolality (*Cesar et al., 2020*). In addition, recent studies have shown that evolution can substantially affect the dynamics of human gut microbiotas (*Garud et al., 2019*; *Yaffe and Relman, 2020*; *Zhao et al., 2019*). It will therefore be illuminating to incorporate evolutionary dynamics into CR models under fluctuating environments (*Good et al., 2018*). Such extended models can then be applied to probe the underlying mechanisms in microbiotas for which frequent sampling and deeper understanding could be translated to urgent applications, including those in marine environments, wastewater treatment plants, and the guts of insect pests and livestock.

## Materials and methods

### Simulations of a CR model with fluctuating resource amounts

Under a serial dilution scheme, an ecological steady state is reached when the dynamics in subsequent passages are identical, which is the case when all consumers are either extinct or have a growth ratio (the ratio of a consumer's final and initial abundances within one passage) equal to the dilution factor $D$. Due to the slow path to extinction of some consumers, reaching an exact ecological steady state can require hundreds of passages, presumably more than realistically occurs between sampling times in the data sets examined here. Thus, we assumed instead that between sampling times the system only approximately reaches an ecological steady state, defined as the growth ratios of all species changing by less than a threshold between subsequent passages that was defined as a fraction of $D$. Throughout this study, $D$ was set to 200 and the steady state threshold was 5%, under which a steady state was approximately reached in about 5 dilutions (*Figure 1B*). In this manner, our model assigns a well-defined state of consumer abundances to each resource environment while ensuring that only a reasonable amount of change occurs between sampling times. Note that in human gut microbiotas, abundances can change by more than 1000-fold between daily samplings (*Figure 2B*), indicating that at least 10 generations can occur between sampling times. The precise value of $D$ did not affect time series statistics, and steady-state thresholds between 1% and 10% generated similar time series statistics (*Figure 1—figure supplement 1*). We therefore expect our results to be robust to the values of these two parameters. Simulations were carried out in Matlab, and all code is freely available online in Matlab and Python at https://bitbucket.org/kchuanglab/consumer-resource-model-for-microbiota-fluctuations/.

### CR model with saturation kinetics

Saturation kinetics were implemented into the CR dynamics of *Equation 1* as

$$\frac{dX_i}{dt} = X_i \left( \sum_{j=1}^{M} R_{ij} \frac{Y_j}{Y_s + Y_j} \right),$$

$$\frac{dY_j}{dt} = -\frac{Y_j}{Y_s + Y_j} \left( \sum_{i=1}^{N} R_{ij} X_i \right),$$

where $Y_s$ denotes the saturation constant. For simplicity, $Y_s$ was assumed to be equal for all resources, and set to an intermediate value of $Y_s = \langle Y_{j,0} \rangle /3$ such that both saturated and linear kinetics could affect community dynamics. Other model details are the same as the original CR model.

### Lotka-Volterra models

The gLV model in *Equation 3* was parametrized in two ways. The first parametrization, which we refer to as CR-converted gLV models, was motivated by the successful recapitulation of experimental time series statistics with our CR model. The CR model can be rewritten as a gLV model when resource consumption rates are similar to the mean value (*Good et al., 2018*). Under this assumption, the mapping is $r_i = 2 \sum_j R_{ij} Y_{j,0}$ and $A_{ij} = \frac{1}{R_{max}} \sum_k R_{ik} R_{jk} Y_{k,0}$. The converted interaction coefficients are negative and symmetric, and their magnitudes depend on the niche overlap between the interacting taxa. Since the resource levels $Y_{j,0}$ are involved in this parametrization, fluctuations in $Y_{j,0}$ across sampling times $T$ translate into fluctuations in $r_i$ and $A_{ij}$.

In the second parametrization, $r_i$ and $A_{ij}$ were randomly drawn from normal distributions with means and variances equal to those in the CR-converted gLV model. $A_{ij}$ were forced to be negative and symmetric.

The gLV models were initialized with equal relative abundances for all taxa, and simulated for a fixed amount of time such that a similar range of relative abundances was generated as in the CR model at approximate ecological steady state.

### Analysis of 16S amplicon sequencing data

Raw 16S sequencing data from *David et al., 2014*; *Song et al., 2020*, were downloaded from the European Nucleotide Archive and the Sequence Read Archive, respectively, and ASVs were extracted using DADA2 (*Callahan et al., 2016*) with default parameters. OTUs or ASVs from other studies were

downloaded and analyzed in their available form. All code for data processing is available in the repository listed above.

## Acknowledgements

We thank members of the Huang lab and Lisa Maier, Rui Fang, Jie Lin, and Felix Wong for helpful discussions. We thank Stephanie Song and Nicholas Chia for sharing metadata. This work was funded by a Stanford School of Medicine Dean's Postdoctoral Fellowship (to PH), NIH F32 GM143859-01 (to PH), an Alfred P Sloan Research Fellowship FG-2021-15708 (to BHG), a Stanford Terman Fellowship (to BHG), NSF grant EF-2125383 (to KCH), NIH Award R01 AI147023 (to KCH), and NIH Award RM1 GM135102 (to KCH). KCH and BHG are Chan Zuckerberg Biohub Investigators.

## Additional information

### Funding

| Funder | Grant reference number | Author |
| --- | --- | --- |
| National Institutes of Health | F32 GM143859-01 | Po-Yi Ho |
| National Institutes of Health | R01 AI147023 | Kerwyn Casey Huang |
| National Institutes of Health | NIH RM1 GM135102 | Kerwyn Casey Huang |
| Alfred P. Sloan Foundation | FG-2021-15708 | Benjamin H Good |
| National Science Foundation | EF-2125383 | Kerwyn Casey Huang |

The funders had no role in study design, data collection and interpretation, or the decision to submit the work for publication.

### Author contributions

Po-Yi Ho, Conceptualization, Data curation, Formal analysis, Funding acquisition, Investigation, Methodology, Resources, Software, Supervision, Validation, Visualization, Writing – original draft, Writing – review and editing; Benjamin H Good, Data curation, Formal analysis, Funding acquisition, Investigation, Methodology, Project administration, Supervision, Validation, Visualization, Writing – original draft, Writing – review and editing; Kerwyn Casey Huang, Conceptualization, Formal analysis, Funding acquisition, Investigation, Methodology, Project administration, Resources, Supervision, Visualization, Writing – original draft, Writing – review and editing

### Author ORCIDs

Benjamin H Good http://orcid.org/0000-0002-7757-3347
Kerwyn Casey Huang http://orcid.org/0000-0002-8043-8138

### Decision letter and Author response

Decision letter https://doi.org/10.7554/eLife.75168.sa1
Author response https://doi.org/10.7554/eLife.75168.sa2

## Additional files

### Supplementary files

• Transparent reporting form

### Data availability

The current manuscript is a computational study, so no data have been generated for this manuscript. Modelling code is uploaded at https://bitbucket.org/kchuanglab/consumer-resource-model-for-microbiota-fluctuations/.

The following previously published datasets were used:

| Author(s) | Year | Dataset title | Dataset URL | Database and Identifier |
|---|---|---|---|---|
| David LA, Materna AC, Friedman J, Campos-Baptista MI, Blackburn MC, Perrotta A, Erdman SE, Alm EJ | 2014 | Host lifestyle affects human microbiota on daily timescales | https://www.ebi.ac.uk/metagenomics/studies/ERP006059 | EBI, ERP006059 |
| Song SD, Acharya KD, Zhu JE, Deveney CM, Walther-Antonio MRS, Tetel MJ, Chia N | 2020 | Daily Vaginal Microbiota Fluctuations Associated with Natural Hormonal Cycle, Contraceptives, Diet, and Exercise | https://www.ncbi.nlm.nih.gov/bioproject/PRJNA637322 | NCBI BioProject, PRJNA637322 |

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
