## [Editor Report]

This paper introduces an elegant mathematical and ecological framework to model the fluctuations of microbial abundances in microbiomes along time series. The modeling approach considers consumer-resource properties and is regulated by few parameters. Applied to time-series microbiome data the model suggests the existence of recurrent patterns of microbial dynamics that are quite dependent on resource competition.

---

## [Decision Letter]

**Decision letter after peer review:**

[Editors’ note: the authors submitted for reconsideration following the decision after peer review. What follows is the decision letter after the first round of review.]

Thank you for submitting the paper "Competition for fluctuating resources reproduces statistics of species abundance over time across wide-ranging microbiotas" for consideration by *eLife*. Your article has been reviewed by 2 peer reviewers, and the evaluation has been overseen by a Reviewing Editor and a Senior Editor. The following individuals involved in review of your submission have agreed to reveal their identity: Sean Gibbons (Reviewer #1).

Comments to the Authors:

We are very sorry to share that, after extensive consultation with the reviewers, we have decided that this work will not be considered further for publication by *eLife*. Both reviewers and the editor think that the mathemathical model proposed is of potential great relevance for the field, but despite the elegant formulation and the interesting results fo some of the analyses, quite a significant amout of additional work would be needed to address most of the reviewers' points and be considered for publication in *eLife* (see below). We are sorry to convey this negative decision, as we addressing the points of the reviewers most likely goes beyond the usual effort for a revision at *eLife*. We are, however, open to considering a substantially revised manuscript in the future.

*Reviewer #1:*

The authors propose a simple consumer-resource (CR) model, where the dynamics of microbial communities are governed by fluctuations in external resources and by competition for these resources between taxa. The model is elegant in its simplicity, while also being biologically intuitive and subtly clever in its implementation. The authors show how the model accurately predicts many of the macroecological patterns found in microbiome time series. Unlike many papers I've read that focus on macroecological patterns (with some exceptions), the authors do a great job connecting model parameters to measured properties of microbial ecosystems and show how these parameterizations of real-world ecosystems can provide potential mechanistic insights into the ecology of the system. I really enjoyed reading this manuscript. The writing is clear, as are the formalisms and the analyses. The model provided many expected results, but also revealed some surprising insights. This is a promising approach for generating novel hypotheses for how microbial ecosystems behave. Overall, I think this is a valuable contribution. My only caveat is that many different mechanistic models can be constructed to explain a given phenomenon -- so I suggest the authors remain somewhat humble about whether or not 'fluctuating resources' are the major drivers of these complex dynamics. They might be! The fact that such a simple model makes so many predictions is promising. But in the end, this is just one possible model among many.

Major Strengths/Weaknesses:

1) I like the simplicity of the CR model. More than this, I like the subtlety with which you handled community dynamics. Many prior studies have erroneously treated microbiome time series as if they directly represent growth curves of all the taxa in the system (e.g. fitting LV models to human gut time series). Your method simulates serial dilution and growth of microbial taxa over several cycles to approximate a steady-state community composition for each sample time point. This fits with my biological intuition.

2) One minor weakness in the data processing was that the most resolved taxonomic level that was analyzed was the family level. Why not start with genus-level? Genus-level annotations can usually be estimated from 16S reads. Another question that I had was whether or not the model assumes absolute or relative abundances? I'm guessing absolute, in which case, I found the rarefaction and renormalization of the counts to frequencies to be a slight concern. I'd suggest the authors perform a centered log-ratio (CLR) transform (or some other form of isometric log-ratio transform) on the non-rarified count data, and only remove low-frequency taxa after the transformation. I doubt this will substantially impact the results, but this is considered best practice.

3) The 'origins of distinctive statistical behaviors…' section is really great. The authors do a great job mapping their model parameters to features that can be estimated directly from the empirical time series (i.e. α-div and the β-slope constrain N, M, and S, while δ-l and s-i constrain σ and k). However, I'm not sure I understood your explanation for why low-sparsity leads to a steeper Taylor's Law slope, and how this is essentially equivalent to a competition-free mode. Naively, I'd expect competition to be greater at low sparsity, due to multiple species consuming the same sets of resources.

4) The non-interacting null model is an appropriate null. However, the authors should be humble about whether or not their competition model is capturing the mechanisms driving community dynamics. For example, direct microbe-microbe killing (antimicrobials or type VI secretion systems) is not captured. Host antimicrobials and immune-system interactions aren't captured. Diet is implicitly captured with the nutrient fluctuations. That being said, I think the model is still reasonable and the insights should be fairly robust -- the environmental fluctuations in the model probably capture a lot of this system-scale variance (in a statistical mechanics kind of way -- the averaging together of a lot of different factors giving rise to a predictable statistical outcome).

5) There seem to be two assumptions regarding time in your model. First, I think you need to be operating within a stationary/stable system (i.e. where there's no long-term drift), correct? I think that's fine but wanted to clarify. The second assumption is that you're sampling from a steady-state end-point of fast internal growth dynamics within the system. I think this is an excellent assumption in the human or mouse gut, but you might want to think about the timescales of sampling and microbial growth in the various systems you are sampling. If you are sampling within the timescales of the faster dynamics (e.g. possible for in vitro systems…maybe in the vaginal system?), how would this impact your results? You mention that your k values were between 0.5 and 1.0, suggesting that internal dynamics were faster than sampling timescales. Due to the ecological steady-state assumption of your modeling, would it be possible for your parameters to tell you that dynamics were slower than sampling timescales?

Overall, I think the authors achieve their aims and that their conclusions are supported by their results. This is an elegant and useful modeling framework that should have a sizable impact on the field and provide potential mechanistic insight into existing and future longitudinal microbiome data sets. I found many of the model predictions to be intuitive, and a few to be surprising, which is always a good sweet spot. I'd like to commend the authors on writing a nice manuscript that clearly communicates their results with a set of beautiful and easy-to-read figures.

*Reviewer #2:*

This paper discusses a consumer-resource model, where microbial families are considered consumers and their nutrients are resources. The model is used to simulate microbial abundances over time: batch feeding events allow populations to grow, dilutions in between feeding events reduce populations. Coefficients of the model, such as the number of resources and the rates at which each family can consume them, are fit to data from different microbiomes by comparing summary statistics of simulated and observed time series. Different microbiome time series, e.g. from mice or humans, have different summary statistics. The model can be optimized to simulate time series with summary statistics similar to each of those from different microbiome data sets.

The model is very simple, allowing the reader to easily understand what is going on. This is a strength of the manuscript. The overlap in resources consumed between consumers in this model is revealed as a crucial parameter because it exhibits the most interesting changes when fitting different microbiome data sets. However, in the model there is no trade off between the rate at which a species may consume a resource and the number of resources it can consume. Therefore, the more different nutrients a species can consume, the fitter it will be. It may be interesting to re-evaluate the major results when this assumption is changed.

A weakness of the paper is that it overstates the implications of the theoretical findings. Simulated timelines from the presented model can generate summary statistics that look like those in real data sets. This will also be possible with other models, even simpler ones or more complex ones. The article ought to include a more critical discussion and validation with simpler (e.g. pairwise interaction) or more complex (e.g. saturating growth kinetics) models.

The article is also poorly referenced, e.g. Niehaus et al. 2019 develop a resource driven model for microbial populations (doi.org/10.1038/s41467-019-10062-x), and Momeni et al. 2017 discussed the importance of resource mediated interactions (doi.org/10.7554/*eLife*.25051).

Finally, the article is not very carefully put together. I received two figures labeled as "Figure 1". The methods appear unfinished.

I recommend reducing the amount of fluff terms throughout the manuscript. For example, the sentence from the abstract:

"Our coarse-grained model parametrizes the intrinsic consumer-resource properties of a community using a small number of macroscopic parameters, including the total number of resources, typical resource fluctuations over time, and the average overlap in resource-consumption profiles across species"

would read fine without the ill-defined filler words:

"Our model parametrizes the consumer-resource properties of a community using parameters that include the total number of resources, resource fluctuations over time, and the average overlap in resource-consumption profiles across species."

In my opinion, simplicity and clarity strengthen theoretical papers, increasing their impact.

[Editors’ note: further revisions were suggested prior to acceptance, as described below.]

Thank you for submitting your article "Competition for fluctuating resources reproduces statistics of species abundance over time across wide-ranging microbiotas" for consideration by *eLife*. Your article has been reviewed by 2 peer reviewers, and the evaluation has been overseen by a Reviewing Editor and Wendy Garrett as the Senior Editor. The following individual involved in review of your submission has agreed to reveal their identity: Sean Gibbons (Reviewer #1).

Essential revisions:

The paper has improved with the revision and it meets the standard for publication in *eLife*. However, the paper is rather technical and in some parts there is the risk of misinterpretation or overestimating/over-interpreting the potential of the model. The authors should better highlight the intrinsic limitations and strong assumptions of the model throughout the paper, starting – for example – from the abstract. It is not a problem of the model or the data per se, but it is rather the way it is communicated considering that the large majority of the readership will have different backgrounds and cannot necessarily understand the limitations directly. Thus, we would like to see a revised manuscript addressing these specific issues as soon as possible.

*Reviewer #1:*

The authors have done a commendable job responding to the reviewer comments. The additional analyses and model simulations have greatly strengthened their work. The authors have provided their code in a more accessible format. And, they have made the suggested improvements in how they discuss their results. I have no further concerns or comments.

*Reviewer #2:*

My main concern remains: a simulation of timeseries is presented that has summary statistics as observed in data. Upon revision, based on my comment that this is not special to the model presented, another model is used; this also reproduces summary statistics similar to those from data. This is not a broad impact result and will, with the current narrative, be easily misunderstood by a non-specialist readership.

In my opinion, such timeseries summary statistics offer little insight and have limited biological meaning. Thus, my original opinion has not shifted much.

---

## [Author Response]

[Editors’ note: the authors resubmitted a revised version of the paper for consideration. What follows is the authors’ response to the first round of review.]

Reviewer #1:The authors propose a simple consumer-resource (CR) model, where the dynamics of microbial communities are governed by fluctuations in external resources and by competition for these resources between taxa. The model is elegant in its simplicity, while also being biologically intuitive and subtly clever in its implementation. The authors show how the model accurately predicts many of the macroecological patterns found in microbiome time series. Unlike many papers I've read that focus on macroecological patterns (with some exceptions), the authors do a great job connecting model parameters to measured properties of microbial ecosystems and show how these parameterizations of real-world ecosystems can provide potential mechanistic insights into the ecology of the system. I really enjoyed reading this manuscript. The writing is clear, as are the formalisms and the analyses. The model provided many expected results, but also revealed some surprising insights. This is a promising approach for generating novel hypotheses for how microbial ecosystems behave.

We thank the reviewer for a careful reading of our manuscript and appreciate the reviewer’s support!

Overall, I think this is a valuable contribution. My only caveat is that many different mechanistic models can be constructed to explain a given phenomenon -- so I suggest the authors remain somewhat humble about whether or not 'fluctuating resources' are the major drivers of these complex dynamics. They might be! The fact that such a simple model makes so many predictions is promising. But in the end, this is just one possible model among many.

We agree with the reviewer that the core of our work is an existence proof, which does not rule out the possibility that other models can also capture experimental data. We have edited the text throughout to better reflect this point.

Moreover, to further explore other models, we have added extensive new simulations of (1) a consumer-resource model with metabolic trade-offs, (2) a consumer-resource model with saturation kinetics, and (3) generalized Lotka-Volterra (gLV) models involving pairwise interactions. It is challenging to exhaustively analyze any particular modeling framework due to the high dimensionality of parameter space, particularly for gLV models that have many more interaction parameters than our macroscopically parametrized consumer-resource (CR) model. To overcome this obstacle, we exploited the fact that our CR model establishes the existence of a simple model that can recapitulate the statistics in microbiota time series, and analyzed the behavior of other models near the parameter space occupied by our successful model. While this approach cannot rule out the existence of other parameter regimes that recapitulate timeseries statistics for other models, we show that it can nevertheless shed light on the features of other models, such as interspecies interactions in LV models, that are necessary to explain the observed statistics. Our approach also highlights some of the challenges that other models may face in describing experimental data. We describe the results for each of modeling framework below in response to reviewer #2, who had similar concerns. We have also revised the text and added several supplemental figures (Figure S10, S11, S13, S14) to incorporate these analyses.

Major Strengths/Weaknesses:(1) I like the simplicity of the CR model. More than this, I like the subtlety with which you handled community dynamics. Many prior studies have erroneously treated microbiome time series as if they directly represent growth curves of all the taxa in the system (e.g. fitting LV models to human gut time series). Your method simulates serial dilution and growth of microbial taxa over several cycles to approximate a steady-state community composition for each sample time point. This fits with my biological intuition.

Thank you! We are glad that the reviewer found the model to be biologically intuitive.

(2) One weakness in the data processing was that the most resolved taxonomic level that was analyzed was the family level. Why not start with genus-level? Genus-level annotations can usually be estimated from 16S reads.

We apologize for the confusion; genus-level annotations can indeed be obtained for the data sets analyzed, but we focused on the family level because it has been suggested to be an appropriate coarse-graining of metabolic capabilities. For instance, prior work found that for diverse soil communities grown in simple medium, family level abundances converged despite substantial variability within families (Goldford *et al. Science* 2018). We therefore reasoned that the family level would be a natural coarse-graining resolution for comparison to a CR model.

Nonetheless, our model and analysis can straightforwardly be applied to other taxonomic levels, and in the original submission, we included an analysis at the class level that was qualitatively similar as the family level (Figure S3 of this revision). Comprehensively simulating systems with hundreds of taxa (e.g. the hundreds to thousands of species found in some gut and soil microbiotas) would require extensive computation time, not to mention the likelihood of metabolic correlations between species in the same genus. To balance these points with the reviewer’s question, in our revision we included an analysis of the gut microbiota time-series data set from Caporaso *et al.* (as in Figure 2) at the genus level. This analysis (Figure 2—figure supplement 2) showed that our CR model could again largely recapitulate all experimental statistics at the genus level. The only statistic that showed a discrepancy was the dominance of the *Bacteroides* genus that disrupted the rank distribution of mean abundances, and recalculation of relative abundances without the *Bacteroides* restored close agreement with model predictions, providing further support that our model can be used at various taxonomic levels.

Another question that I had was whether or not the model assumes absolute or relative abundances? I'm guessing absolute, in which case, I found the rarefaction and renormalization of the counts to frequencies to be a slight concern. I'd suggest the authors perform a centered log-ratio (CLR) transform (or some other form of isometric log-ratio transform) on the non-rarified count data, and only remove low-frequency taxa after the transformation. I doubt this will substantially impact the results, but this is considered best practice.

We thank the reviewer for bringing to attention the centered log-ratio transform and its uses in analyzing compositional data. First, we reaffirm that in all our analyses, experimental data and numerical simulations were processed and analyzed equivalently, ensuring the validity of the analyses. Our model was formulated using absolute abundances, which were then always normalized to sum to one before comparing to compositional (relative abundance) experimental data. In fact, since absolute abundances within the model were never analyzed directly, the total resource levels can be normalized to sum to one a priori without affecting the results.

Moreover, given the typical limits of detection of 16S amplicon sequencing data sets, we only examined time series statistics for taxa with relative abundance >10^-4^ at any given time point. Previously, we renormalized the relative abundances after removing taxa below the detection threshold. The results without renormalization were unchanged (Author response image 1) , which is intuitive because taxa below the detection threshold comprise only a tiny fraction of the community.

**Author response image 1. sa2fig1:** Taxa below the limit of detection do not affect time series statistics. Shown are data from Caporaso *et al.* as in Figure 2. The original analysis is shown as dotted black lines. Colored lines denote model predictions without renormalizing the relative abundances after ignoring taxa with relative abundance below the detectability threshold of 10-4; the two lines are virtually indistinguishable in every case.

Finally, the denominator in the CLR is the geometric mean, which cannot naturally handle cases with zero reads. It is therefore not a natural metric to describe the time-series statistics of lowabundance taxa that are fluctuating above and below the limit of detection. On the other hand, relative abundances naturally incorporate cases with zero reads. Since we processed and analyzed experimental and simulated data equivalently, statistics such as the residence and return times can be compared consistently across samples and between data and simulations.

We have revised the text to reflect these points.

(3) The 'origins of distinctive statistical behaviors…' section is really great. The authors do a great job mapping their model parameters to features that can be estimated directly from the empirical time series (i.e. α-div and the β-slope constrain N, M, and S, while δ-l and s-i constrain σ and k). However, I'm not sure I understood your explanation for why low-sparsity leads to a steeper Taylor's Law slope, and how this is essentially equivalent to a competition-free mode. Naively, I'd expect competition to be greater at low sparsity, due to multiple species consuming the same sets of resources.

We would like to stress that our explanation for the value of the slope in Taylor’s law is partial and does not account for all possible effects, and have edited the text to reflect this point. Our reasoning was that since our model has no trade-offs or metabolic constraints between the number of resources consumed and a consumer’s total consumption rate (see e.g. Posfai *et al. PRL* 2017, Good *et al. PNAS* 2018, and response to reviewer #2), the number of resources consumed necessarily strongly affects consumer abundance. This effect dominates at high sparsity, in which consumers typically consume distinct sets of resources and the number of resources consumed is relatively variable, as depicted in Figure S7A. Indeed, the Taylor’s law slope predicted by this no-competition model (Figure S7A) matched closely with simulations of our model at high sparsity (Figure S7B).

Despite obvious competition in the actual CR model when sparsity is low, we nevertheless attempted to understand the low-sparsity limit by extrapolating the no-competition model to the limiting case of zero sparsity, in which the number of resources consumed is the same for all consumers. Surprisingly, when the mean number of resources consumed is large, the predictions from the no-competition model matched qualitatively with simulations of the actual CR model, suggesting that reduction in the variance of the number of resources consumed partially explains the dependence of Taylor’s law on sparsity. We emphasize that we do not claim that zero sparsity is equivalent to a competition-free model, and we have revised the text to explain this point more clearly.

(4) The non-interacting null model is an appropriate null. However, the authors should be humble about whether or not their competition model is capturing the mechanisms driving community dynamics. For example, direct microbe-microbe killing (antimicrobials or type VI secretion systems) is not captured. Host antimicrobials and immune-system interactions aren't captured. Diet is implicitly captured with the nutrient fluctuations. That being said, I think the model is still reasonable and the insights should be fairly robust -- the environmental fluctuations in the model probably capture a lot of this system-scale variance (in a statistical mechanics kind of way -- the averaging together of a lot of different factors giving rise to a predictable statistical outcome).

We agree with the reviewer that there exist other mechanisms that might affect community dynamics. We have revised the text to emphasize that our model is an existence proof and does not rule out that other mechanisms might drive community dynamics.

In addition, we tested the robustness of the macroscopic parameters of our model by considering variants incorporating saturation kinetics and metabolic trade-offs (see response to reviewer #2). Despite these modifications, the best-fit parameters for the original model still reproduced data under mild assumptions, suggesting that the insights obtained are robust to model details to a reasonable extent. We have revised the text to include this discussion.

(5) There seem to be two assumptions regarding time in your model. First, I think you need to be operating within a stationary/stable system (i.e. where there's no long-term drift), correct? I think that's fine but wanted to clarify. The second assumption is that you're sampling from a steady-state end-point of fast internal growth dynamics within the system. I think this is an excellent assumption in the human or mouse gut, but you might want to think about the timescales of sampling and microbial growth in the various systems you are sampling. If you are sampling within the timescales of the faster dynamics (e.g. possible for in vitro systems…maybe in the vaginal system?), how would this impact your results? You mention that your k values were between 0.5 and 1.0, suggesting that internal dynamics were faster than sampling timescales. Due to the ecological steady-state assumption of your modeling, would it be possible for your parameters to tell you that dynamics were slower than sampling timescales?

Yes, we assume long-term stability without drift, which we now clarify in the text.

We were indeed motivated by gut microbiotas when we assumed that the internal time scales between samplings were faster, and we agree that this assumption may not be the case for all systems analyzed here. In particular, the in vitro system was sampled every log_!_ 200 ≈ 7.6 generations, and hence may not have reached ecological steady state between samplings. Our model nevertheless produced a good fit (Figure S20), indicating that model results were robust to a relatively broad range of internal time scales (see Figure S1 and the discussion below).

More systematically, three factors control the relationship between the internal and sampling time scales: the dilution factor, the threshold change for ecological steady state, and the reservoir composition. The dilution factor and threshold affect the number of generations between samplings, while reservoir composition affects the correlation between sampling times. Simulation results were not substantially affected for a relatively broad range of dilution factors and thresholds (Figure S1). On the other hand, if the reservoir inherits a substantial fraction of its composition from the previous sampling time, simulation results can be affected (Figure S2). These results show that the relationship between the internal and sampling time scales can affect time-series statistics in complex ways. It remains an interesting open question to infer internal time scales from microbiota time series, and our work provides a strong starting point to do so. We have revised the text to incorporate this discussion.

Overall, I think the authors achieve their aims and that their conclusions are supported by their results. This is an elegant and useful modeling framework that should have a sizable impact on the field and provide potential mechanistic insight into existing and future longitudinal microbiome data sets. I found many of the model predictions to be intuitive, and a few to be surprising, which is always a good sweet spot. I'd like to commend the authors on writing a nice manuscript that clearly communicates their results with a set of beautiful and easy-to-read figures.

We thank the reviewer for their kind words and helpful review.

Reviewer #2:This paper discusses a consumer-resource model, where microbial families are considered consumers and their nutrients are resources. The model is used to simulate microbial abundances over time: batch feeding events allow populations to grow, dilutions in between feeding events reduce populations. Coefficients of the model, such as the number of resources and the rates at which each family can consume them, are fit to data from different microbiomes by comparing summary statistics of simulated and observed time series. Different microbiome time series, e.g. from mice or humans, have different summary statistics. The model can be optimized to simulate time series with summary statistics similar to each of those from different microbiome data sets.The model is very simple, allowing the reader to easily understand what is going on. This is a strength of the manuscript. The overlap in resources consumed between consumers in this model is revealed as a crucial parameter because it exhibits the most interesting changes when fitting different microbiome data sets.

We thank the reviewer for their careful reading of our manuscript, and appreciate the reviewer’s support for the strength of our work.

However, in the model there is no trade off between the rate at which a species may consume a resource and the number of resources it can consume. Therefore, the more different nutrients a species can consume, the fitter it will be. It may be interesting to re-evaluate the major results when this assumption is changed.

We appreciate the reviewer’s point and note indeed that metabolic trade-offs have been investigated previously in other contexts (e.g. Posfai *et al. PRL* 2017, Tikhonov and Monasson *PRL* 2017, Good *et al. PNAS* 2018). To explore how metabolic trade-offs affect time-series statistics, we simulated our original model with the constraint that the sum of consumption rates ∑jRij for consumer і is normalized to a fixed capacity R~i. We further assumed R~i to be randomly drawn from the original growth rates ∑jRijYj,0 (consumption rate times resource level), in effect preserving the variation in consumer fitness while removing its correlation to the number of resources consumed. Interestingly, this model largely reproduced all the time-series statistics using the same best-fit parameters of the original model, indicating that the correlation between fitness and the number of resources in the original model is not required to reproduce experimental data (Figure S10 of the revision). We have revised the text to incorporate this finding.

A weakness of the paper is that it overstates the implications of the theoretical findings. Simulated timelines from the presented model can generate summary statistics that look like those in real data sets. This will also be possible with other models, even simpler ones or more complex ones. The article ought to include a more critical discussion and validation with simpler (e.g. pairwise interaction) or more complex (e.g. saturating growth kinetics) models.

We appreciate the reviewer’s point and have now taken more care not to overstate the implications of our findings. We now emphasize in the text that the core of our work is an existence proof of a model that can recapitulate the statistics of experimental time series, and that our work does not rule out the possibility that other models can also capture experimental statistics.

Moreover, we have endeavored to directly address the reviewer’s points about pairwise interactions in the form of generalized Lotka Volterra (gLV) models and the original CR model with saturating kinetics, as described below and in the text.

Generalized Lotka-Volterra models: In the gLV model, Ν taxa grow and interact via dXidt=Xi(ri+∑j=iNAijXj) where ✕_"_ denotes the abundance of taxon і, r_"_ its growth rate, and *A_ij_* its interaction coefficient with taxon j. Since this classical model is generally unstable for randomly drawn interaction coefficients (May *Nature* 1972), we focused on instances of the gLV model near the parameter space corresponding to the dynamics of our CR model. This conversion between models was achieved by converting the consumption rates *R_ij_* and resource levels *Y_j,0_* at each sampling time T to the growth rates r_"_ and interaction coefficients A_"#_ that characterize the dynamics when consumption rates are similar to the mean value (Good *et al. PNAS* 2018). Under this assumption, the mapping is ri=2∑jRijYj,o and Aij=1Rmax∑kRikRjkYk,0 , which we refer to as CR-converted cLV models. The converted interaction coefficients are negative and symmetric, and their magnitudes depend on the niche overlap between the interacting taxa. Since the resource levels *Y_j,_*_0_ are involved in this parameterization, fluctuations in Y_#,%_ across sampling times T translate into fluctuations in *r_i_* and *A_ij_*.

Finally, since the data being examined is compositional, the gLV model was amended to describe only the dynamics of relative abundances (Joseph *et al. PLoS Comp Biol* 2020) by including a normalizing term Γ(t),

dXidt=Xi(ri+∑j=iNAijXj−Γ(t))where ✕*_i_* now describes the relative abundance of taxon і and Γ(t) = ∑*_i_* r*_i_*✕_*i*_ + ∑_*I,j*_ A*_ij_*✕*_i_*✕_*j*._ Relative abundances were initialized with equal values across all taxa, and the normalizing term Γ(t) enforces ∑_*i*_ ✕*_j_* = 1. The cLV models were simulated for a fixed amount of time such that a similar range of relative abundances is generated as in the CR model at approximate ecological steady state. These CR-converted compositional Lotka-Volterra models generated time series statistics that reproduced the experimental data to a similar extent as the original CR model (Figure S13A). Similarly, the CR-converted cLV models better predicted the experimentally observed distribution of pairwise correlations compared with the non-interacting null model (Figure S13B).

Finally, we asked whether more general ensembles of r*_i_* and A*_ij_* could also reproduce the experimental data. We randomly selected r*_i_* and A*_ij_* from normal distributions with means and variances equal to those in the CR-converted cLV models while enforcing symmetric and negative interactions. The resulting cLV models yielded a poor fit to the data (Figure S14). Together, these results suggest that pairwise interactions between taxa are likely sufficient to recapitulate the experimental data, although their parameters must be drawn from particular statistical ensembles.

Saturating-kinetics model: The CR model with saturating kinetics is the same as the original CR model, except that the dynamical equations are now dXidt=Xi(∑j=1MRijYjYs+Yj)dYjdt=−YjYs+Yj(∑i=1NRijXi) where Ys denotes the saturation constant. For simplicity, Ys was assumed to be equal for all resources, and set to an intermediate value of Ys=⟨Yj,0⟩/3 such that both saturated and linear kinetics could affect community dynamics. When this model was simulated with the best-fit parameters of the original model, the resulting dynamics were much less variable across sampling times than without saturating kinetics (Figure S11). Intuitively, this result is because the saturated regime is unaffected by small changes in resource levels.

Indeed, experimental statistics were again reproduced after the strength of environmental fluctuations σ was increased.

The above results demonstrate how our approach can reveal the features of other models that are necessary (or not) to explain experimental data. We have added the above figures and discussions to the text, expanding the context for our CR model through comparisons with these other models.

The article is also poorly referenced, e.g. Niehaus et al. 2019 develop a resource driven model for microbial populations (doi.org/10.1038/s41467-019-10062-x), and Momeni et al. 2017 discussed the importance of resource mediated interactions (doi.org/10.7554/eLife.25051).

We agree that these papers are important to include and apologize for our oversight. They are now cited appropriately.

Finally, the article is not very carefully put together. I received two figures labeled as "Figure 1". The methods appear unfinished.

We thank the reviewer for their attention to detail. We have carefully combed through the manuscript to ensure that all figures, citations, and references are correct. The Methods were in fact complete, but we have expanded the section for clarity and to reflect the addition of several models.

I recommend reducing the amount of fluff terms throughout the manuscript. For example, the sentence from the abstract:"Our coarse-grained model parametrizes the intrinsic consumer-resource properties of a community using a small number of macroscopic parameters, including the total number of resources, typical resource fluctuations over time, and the average overlap in resource-consumption profiles across species"would read fine without the ill-defined filler words:"Our model parametrizes the consumer-resource properties of a community using parameters that include the total number of resources, resource fluctuations over time, and the average overlap in resource-consumption profiles across species."In my opinion, simplicity and clarity strengthen theoretical papers, increasing their impact.

We appreciate the reviewer’s point and have edited the text for conciseness throughout. We have edited that particular sentence as suggested but left in the adjective “coarse-grained” as we feel it is important to convey to the reader that the “resources” do not necessarily correspond to individual metabolites.

[Editors’ note: what follows is the authors’ response to the second round of review.]

Essential revisions:The paper has improved with the revision and it meets the standard for publication in eLife. However, the paper is rather technical and in some parts there is the risk of misinterpretation or overestimating/over-interpreting the potential of the model. The authors should better highlight the intrinsic limitations and strong assumptions of the model throughout the paper, starting – for example – from the abstract. It is not a problem of the model or the data per se, but it is rather the way it is communicated considering that the large majority of the readership will have different backgrounds and cannot necessarily understand the limitations directly. Thus, we would like to see a revised manuscript addressing these specific issues as soon as possible.Reviewer #2:My main concern remains: a simulation of timeseries is presented that has summary statistics as observed in data. Upon revision, based on my comment that this is not special to the model presented, another model is used; this also reproduces summary statistics similar to those from data. This is not a broad impact result and will, with the current narrative, be easily misunderstood by a non-specialist readership.

Our key finding in response to the reviewer’s previous comment was the following: Although the generalized Lotka-Volterra (gLV) model can also reproduce experimental statistics, its parametrization was guided by the CR model. The guidance provided by the consumer-resource (CR) model was crucial because null parametrizations of GLV models that one might use, e.g., normally distributed interaction coefficients, did not reproduce experimental statistics. These results were shown in Figure S13 and S14. In other words, without having first identified the CR models that reproduce data, it would be highly unlikely to find the mathematically related gLV models that also reproduce data. This finding therefore strengthens the implications of our modeling framework, which can aid the investigation of other ecological models.

We apologize for not stating our findings more clearly and have revised the text throughout for clarity and to avoid mis-interpretation. The extensive changes can be found highlighted in the “highlighted” version of the manuscript file.

In my opinion, such timeseries summary statistics offer little insight and have limited biological meaning. Thus, my original opinion has not shifted much.

A growing body of work has begun to show that time series statistics can be a useful window into the difficult-to-access inner workings of complex microbiotas. We highlight some important results from this body of work and clarified their implications in the introduction. In particular only subsets of models can reproduce experimental statistics, implying that these time series statistics are informative of the underlying dynamics. Our work extends the variety of existing insights garnered from time series statistics and offers a baseline parametrization of CR models for complex microbiotas. Thus, we believe that these results have broad applications, as we elaborate on in the discussion.